# When Should Lymphadenectomy Be Performed in Non-Metastatic Pancreatic Neuroendocrine Tumors? A Population-Based Analysis of the German Clinical Cancer Registry Group

**DOI:** 10.3390/cancers16020440

**Published:** 2024-01-19

**Authors:** Thaer S. A. Abdalla, Louisa Bolm, Monika Klinkhammer-Schalke, Sylke Ruth Zeissig, Kees Kleihues van Tol, Peter Bronsert, Stanislav Litkevych, Kim C. Honselmann, Rüdiger Braun, Judith Gebauer, Richard Hummel, Tobias Keck, Ulrich Friedrich Wellner, Steffen Deichmann

**Affiliations:** 1Department of Surgery, University Medical Center Schleswig-Holstein, Campus Lübeck, 23562 Lübeck, Germany; 2Network for Care Quality and Research in Oncology (ADT), German Cancer Registry Group of the Society of German Tumor Centers, 14057 Berlin, Germany; 3Institute of Clinical Epidemiology and Biometry (ICE-B), University of Würzburg, 97070 Würzburg, Germany; 4Department of Pathology, University Medical Center Freiburg, 79106 Freiburg, Germany; 5Department of Internal Medicine I, University Medical Center Schleswig-Holstein, Campus Lübeck, 23562 Lübeck, Germany; 6Department of Surgery, University Medical Center Greifswald, 17475 Greifswald, Germany

**Keywords:** pancreatic neuroendocrine tumors (pNETs), lymph node metastasis, lymphadenectomy, population-based analysis

## Abstract

**Simple Summary:**

This population-based study of 1006 individuals diagnosed with non-metastatic pancreatic neuroendocrine tumors (pNETs) aimed to assess the significance of lymph node metastasis (LNM). The presence of LNM emerged as an independent prognostic factor associated with a reduced disease-free survival (DFS) in the multivariable analysis. Remarkably, LNM was identified in nearly 25% of surgically resected pNET cases, highlighting the pivotal role of lymphadenectomy in the management of pNETs. In conclusion, this study culminated in the development of a predictive model integrating variables linked to LNM. This model serves as a valuable tool for the preoperative identification of patients at risk of LNM, offering insights that can guide clinical decision-making and enhance patient care in the context of pNETs.

**Abstract:**

Background: Patient selection for lymphadenectomy remains a controversial aspect in the treatment of pancreatic neuroendocrine tumors (pNETs), given the growing importance of parenchyma-sparing resections and minimally invasive procedures. Methods: This population-based analysis was derived from the German Cancer Registry Group during the period from 2000 to 2021. Patients with upfront resected non-functional non-metastatic pNETs were included. Results: Out of 5520 patients with pNET, 1006 patients met the inclusion criteria. Fifty-three percent of the patients were male. The median age was 64 ± 17 years. G1, G2, and G3 pNETs were found in 57%, 37%, and 7% of the patients, respectively. Lymph node metastasis (LNM) was present in 253 (24%) of all patients. LNM was an independent prognostic factor (HR 1.79, CI 95% 1.21–2.64, *p* = 0.001) for disease-free survival (DFS). The 3-, 5-, and 10-year disease-free survival in nodal negative tumors compared to nodal positive was 82% vs. 53%, 75% vs. 38%, and 48% vs. 16%. LNM was present in 5% of T1 tumors, 25% of T2 tumors, and 49% of T3–T4 tumors. In T1 tumors, G1 was the most predominant tumor grade (80%). However, in T2 tumors, G2 and G3 represented 44% and 5% of all tumors. LNM was associated with tumors located in the pancreatic head (*p* < 0.001), positive resection margin (*p* < 0.001), tumors larger than 2 cm (*p* < 0.001), and higher tumor grade (*p* < 0.001). The multivariable analysis showed that tumor size, tumor grade, and location were independent prognostic factors associated with LNM that could potentially be used to predict LNM preoperatively. Conclusion: LNM is an independent negative prognostic factor for DFS in pNETs. Due to the low incidence of LNM in T1 tumors (5%), parenchyma-sparing surgery seems oncologically adequate in small G1 pNETs, while regional lymphadenectomy should be recommended in T2 or G2/G3 pNETs.

## 1. Introduction

Pancreatic neuroendocrine tumors (pNETs) are a rare entity constituting about 2–4% of all pancreatic neoplasms [1,2]. Due to the increased use of cross-sectional imaging among others, the incidence of pNETs has increased five-fold over the past three decades based on data from the surveillance, epidemiology, and end results (SEER) database [3]. pNETs can be divided into functional and non-functional pNETs according to their secretion of hormones and the presence of hormone-related symptoms [4]. Non-functional pNETs (NF-pNETs) account for 60–70% of all pNETs [5]. These tumors are hormonally inactive and may be associated with non-specific symptoms because of the tumor mass effect. These tumors are often detected incidentally in radiologic studies performed for other unrelated reasons. Due to delayed diagnosis, NF-pNETs can be associated with locoregional and metastatic tumor spread and therefore with worse prognosis [6].

While tumor resection is always advocated in the case of functional pNETs [7,8], the current guidelines have been inconsistent on indications for surgery versus watch-and-wait, the extent of parenchymal resection, as well as when to perform radical lymphadenectomy in non-functional pNETs [7,9,10]. A watch-and-wait strategy has been suggested as an alternative for small (<2 cm) and well-differentiated tumors [7]. The scarcely available literature on the prognostic impact of lymph node metastasis in pNETs and its predictors as well as the unclear relation of tumor size to lymph node metastasis and tumor grade has led to variable recommendations among international guidelines and therefore variable practice among surgeons.

This study aimed to identify the preoperative factors associated with lymph node metastasis in pNETs without distant metastasis and to evaluate their impact on survival, using a large-scale dataset from the German Clinical Cancer Registry Group.

## 2. Materials and Methods

### 2.1. Study Design and Database

Conducted by the German Cancer Registry Group (GCRG) of the Society of German Tumor Centers (ADT), this investigation utilized anonymized data extracted from 20 clinical registries spanning the years 2000 to 2021, adhering to ADT regulations. Ethical approval for the protocol was obtained from the local ethics committee of the University of Lübeck, Germany (Approval No: #2023-156).

Within the cohort of patients diagnosed with pancreatic malignancy (ICD-O 3rd edition codes C25.x), those with non-functional pancreatic neuroendocrine tumors (pNETs) (ICD-O-3 morphology codes 8240-1/3, 8246, and 8249/3) were specifically isolated for analysis. Exclusion criteria encompassed patients with functional pNETs such as Insulinomas, Glucagonomas, Gastrinomas, VIPomas, Somatostatinomas, as well as mixed neuroendocrine non-neuroendocrine neoplasia and Islet cell carcinoma (ICD-O-3 8150-3, 8151-3, 8152-3, 8153-3, 8154-3, 8155-3, and 8156-3) [11]. The study focused exclusively on patients without distant metastasis who underwent initial surgical resection.

Data extracted from the registry included patient demographics (sex, age at diagnosis), disease characteristics (lymph node metastases, T-stage, lymphangiosis, hemangiosis, grading), treatment details (resection status, tumor location, type of therapy, operation type), and follow-up metrics (follow-up time, status at last follow-up). Certain variables, namely age, lymph node metastasis, tumor location, and resection status, were dichotomized for analysis (age: ≤65 years vs. >65 years; lymph node metastasis: N0 vs. LNM; resection status: R0 vs. R+). Due to changes in the TNM classification over the study period, T-stage categories T3 and T4 were combined, as retrospective restaging without metric data on tumor size was unfeasible.

### 2.2. Statistical Methods

For statistical analysis, SPSS 28 for Windows (IBM, Armonk, NY, USA) was used. Descriptive statistics were used to describe patient baseline characteristics. To evaluate categorical variables, the Chi-squared test was applied. Survival curves were plotted using the Kaplan–Meier method and compared using the log-rank test. Results are presented as median survival in months.

The overall survival was defined as the period from the date of diagnosis to either the date of death or the last follow-up, whichever occurred first. The disease-free survival was defined as the period from the date of operation to the date of disease recurrence or last follow-up, whichever occurred first. Uni- and multivariable Cox regression analyses were used to determine prognostic variables for survival. All statistical tests were performed two-sided with a significance level of *p* = 0.05.

## 3. Results

### 3.1. Patient Cohort

Based on the clinical cancer registry data, 5520 patients with pNETs were identified, from which 1006 patients met the inclusion and exclusion criteria and were included in this study (Appendix A). There were slightly more male patients compared to female patients (53% vs. 47%). T1 tumors were present in 404 (40%) of the patients, T2 tumors in 289 (28%), and T3–T4 tumors in 313 (32%). The median age at diagnosis was 64 ± 17 years. Tumor location was documented in 780 of the patients. The most common location was the pancreatic head (n = 313, 40%), followed by the pancreatic tail (n = 307, 39%) and pancreatic body (n = 160, 21%). Pancreatic head resection was performed in 300 (37%) patients, distal pancreatectomy in 478 (58%) patients, and total pancreatectomy in 44 (4%) patients. In 184 patients, pancreatic resection was conducted; however, the exact type was not documented. Patient characteristics and operative information are shown in Table 1.

### 3.2. Impact of LNM on Overall Survival and Disease-Free Survival

The 3-, 5-, and 10-year overall survival in nodal-negative tumors compared to nodal positive was 84% vs. 79% (*p* = 0.505), 80% vs. 68% (*p* = 0.89), and 56% vs. 52% (*p* = 0.707). Univariable analysis showed that higher age (*p* < 0.001), male sex (*p* = 0.008), tumor grade (*p* < 0.001), positive resection margin (*p* < 0.001), and tumors located in the pancreatic head (*p* < 0.001) were associated with reduced overall survival. Here, LNM did not qualify as a statistically significant prognostic factor.

Information regarding disease-free survival was present in n = 512 patients. The 3-, 5-, and 10-year disease-free survival in nodal negative tumors compared to nodal positive was 82% vs. 53 (*p* < 0.001), 75% vs. 38 (*p* < 0.001), and 48% vs. 16% (*p* < 0.001). Univariable analysis revealed that LNM (*p* < 0.001), tumor size (*p* < 0.001) (Appendix A), tumor grade (*p* < 0.001), positive resection margin (*p* < 0.001), and tumor location in the pancreatic head (*p* = 0.003) were associated with shorter disease-free survival. The multivariable analysis showed that tumor size >2 cm (HR 2.76, CI 95% 1.53–5.00, *p* < 0.001), positive resection margins (HR 1,80, CI 95% 1.01–3.23, *p* = 0.045), tumor grade (G2 vs. G1 (HR 2.12, CI 95% 1.38–3.25, *p* < 0.001) and G3 vs. G1 (HR 3.73, CI 95% 2.11–6.53, *p* < 0.001)), and LNM (HR 1.88, CI 95% 1.27–2.78, *p* = 0.001) were independent negative prognostic factors of disease-free survival (Table 2).

### 3.3. Factors Associated with Lymph Node Metastasis

Out of 1006 patients, lymph node metastasis was present in 253 (24%) patients. LNM was present in 5% of patients with T1 tumors, 25% of T2 tumors, and 49% of T3–T4 tumors (Figure 1). Moreover, LNM was present in 13% of G1 tumors, 35% of G2 tumors, and 61% of G3 tumors (Table 3). Tumor grade increased with tumor size: While T1 tumors were mostly G1 in 80%, T2 tumors were mostly G2 (44%) and G3 (5%). Interestingly, the incidence of LNM in T1 was as high as 10% in the case of T1 G2 and 20% in T1 G3 (Figure 2). Statistically, LNM was associated with tumors located in the pancreatic head (*p* = 0.002), positive resection margin (*p* < 0.001), tumors larger than 2 cm (*p* < 0.001), and higher tumor grade (G2 vs. G1 and G3 vs. G1, *p* < 0.001).

Tumor location was associated with LNM in T2 tumors. The incidence of LNM in T2 pNETs was as high as 31% when located in the pancreatic head as compared to 18% when located in the pancreatic body and tail (Figure 3). However, in T1 tumors, the incidence of LNM was similar and did not reach statistical significance in either location (6% vs. 3%, respectively) due to low case numbers.

Multivariable analysis demonstrated that tumor location in the pancreatic head (HR 1.87, 95%-CI 1.26–2.77, *p* < 0.001), tumor grade G2 vs. G1 (HR 2.23, 95%-CI 1.45–3.43, *p* < 0.001), G3 vs. G1 (HR 3.65, 95%-CI 1.79–7.32, *p* < 0.001), as well as T2 and T3–T4 compared to the T1 tumor stage (HR 5.31, 95%-CI 2.69–10.59, *p* < 0.001 and HR 14.84, 95%-CI 7.67–28.72, *p* < 0.001) were independently associated with LNM (Table 4).

### 3.4. Preoperative Prediction of Lymph Node Metastasis

In multivariable analysis, a significant association could be demonstrated between LNM and tumor size, tumor grading, and tumor location. These three parameters can potentially be identified prior to the surgical procedure by cross-sectional imaging and biopsy and can offer valuable information regarding N+. We created a prediction model with the aim of using these preoperative identifiable parameters, which, however, originated from the postoperative histology in our cohort. In this subgroup analysis, we included 780 patients with complete information regarding tumor size, tumor grading, and tumor location. Therefore, prediction models using these parameters were evaluated, including size alone, tumor size and location, as well as tumor size, location, and grade as predictors (Figure 4). Combined tumor size, grade, and location provided the most accurate prediction of LNM (AUC 0.832, 95%-CI 0.80–0.86, *p* < 0.001). The results of the prediction model are illustrated in Appendix A.

### 3.5. Clinical Versus Histopathological Nodal Stage

Information regarding the preoperative clinical lymph node stage (cN) was present in 156 patients. Out of 109 patients with the cN0 stage, 14 patients had pN1 status. Furthermore, out of 34 patients with the cN1 stage, 12 patients had no LNM (pN0). The calculated sensitivity, specificity, positive predictive value, and negative predictive value were 61.1%, 90.1%, 64.7%, and 88.6%, respectively. Information on the modality of staging was not available, however.

### 3.6. Relation of Lymph Node Metastasis to Local or Distant Recurrence

The incidence of local and distant recurrence in follow-up after resection was compared between tumors with versus without LNM at the time of resection: In small pNETs (<2 cm), LNMs were not associated with local recurrence (2% vs. 0%, *p* = ns), but with distant metastasis (13% vs. 3%, *p* < 0.031). In pNETs > 2 cm, the incidence of local recurrence (5% vs. 8%, *p*-value = 0.034) as well as distant metastasis (19% vs. 37%, *p*-value < 0.001) was significantly increased in tumors with LNM (Appendix A).

## 4. Discussion

The recommendations for the extent of surgical resection in patients with non-metastatic non-functional pNETs are still not unanimous among the different guidelines. While The European Neuroendocrine Tumor Society (ENETS) guidelines recommend pancreatic resection of pNETs > 2 cm, it does not provide a clear indication for lymphadenectomy or the extent of pancreatic resection [7]. On the other hand, the North American Neuroendocrine Tumor Society (NANETS) consensus guidelines and the National Comprehensive Cancer Network (NCCN) recommend tumor resection with lymphadenectomy for all tumors larger than 2 cm [9,10,12]. The clinical importance of LNM remains controversial as different studies demonstrate heterogeneous results regarding the association of LNM with patients’ survival. This variability might be in part caused by the small number of patients and more importantly, different follow-up times among these studies. To investigate this topic, we analyzed the data delivered from 20 clinical cancer registries in Germany. Here we identified four relevant findings regarding LNM and its oncologic importance and prognosis.

First, the presence of LNM was not associated with overall survival (OS); however, LNM represented an independent prognostic variable for disease-free survival (DFS), which is a more valid parameter considering the indolent biology of pNETs [13]. Over the years, the association of LNM with oncological outcomes has been controversial. Some authors have concluded that nodal positivity negatively impacts survival [14,15], while others have not [16,17,18]. However, a recent meta-analysis reported significantly worse survival in patients with non-functional pNETs [19].

Second, we investigated the incidence of LNM in pNETs. In our cohort, LNM was present only in 5% of T1 tumors but 25% of T2 and 45% of T3–4 tumors. This is in line with previous publications, which describe a 10–14% incidence of LNM in pNETs <2 cm and up to 25% in pNETs larger than 2 cm [14,19,20,21]. Moreover, we observed an exponential shift in the distribution of tumor grade in tumors larger than 2 cm. While T1 tumors were 80% G1, T2 tumors were mostly (53%) G2/G3, which in turn is also a poor prognostic factor for DFS and OS in pNETs [13,22,23].

It is worth comparing these data to early colorectal cancer. An endoscopically resected T1 rectal cancer with high-risk features (high grade, lymphangiosis, mucosal infiltration >1000) should undergo a salvage oncologic resection with TME due to a probability of LNM of 10–20% [24,25,26]. Therefore, why should a 2–4 cm pNET with a probability of LNM of 25% undergo localized resection without lymphadenectomy?

Furthermore, all guidelines recommend oncologic resection and lymphadenectomy in the presence of suspicious lymph nodes in imaging studies [7,9,10]. Some authors recommend parenchyma-sparing resection in tumors <3 cm when imaging studies show nodal-negative disease [15]. However, a recent prospective study demonstrated that current imaging studies like contrast-enhanced CT, endoscopic ultrasonography, and ^68^Ga-DOTATOC PET are not sensitive to the preoperative detection of LNM in sporadic NF-pNETs (sensitivity of 26%, 19%, and 12%, respectively) despite their high specificity (95%, 98%, and 95%) [27]. In our study, information regarding the preoperative clinical nodal stage was available in 156 patients only; here, the sensitivity of clinical nodal staging was low (61%). Although the calculated sensitivity was higher than the latter study, our results support the findings of Partelli et al. in that the current imaging studies have a low sensitivity for the diagnosis of LNM in pNETs. This would not represent a major drawback when considering tumors <2 cm since the incidence of LNM was 5%, but this might be different when looking at tumors larger than 2 cm, which have a higher incidence of LNM of 25% and advanced tumor grades in more than half of the patients.

Thirdly, we identified tumor size, location, and grade as independent prognostic variables for the presence of LNM after resection (Table 4). Since tumor size, tumor location, and tumor grade can potentially be identified preoperatively [28,29,30], we used these parameters for LNM prediction and found the best performance in the combined prediction model (Appendix A). For instance, a T1 G2 pNET situated in the pancreatic head was considered a high-risk situation given its association with lymph node metastasis (LNM) in 18% of cases. Conversely, the identical tumor manifested a comparatively lower risk of LNM at 6% when located in the pancreatic body or tail. This discrepancy underscores the potential consequences of forgoing lymphadenectomy in the former scenario, as it may predispose to the presence of undetected metastatic disease, ultimately contributing to an increased likelihood of early tumor recurrence.

Lastly, we compared patterns of recurrence in patients with nodal-negative vs. positive disease at the time of resection. In small pNETs (<2 cm), we observed a markedly increased incidence of distant metastasis, while in pNETs ≥ 2 cm, the incidence of local as well as distant recurrence was higher. This phenomenon might be explained by the higher incidence of G2/G3 tumors in tumors larger than 2 cm, which relates to lymphatic tumor spread, which leads to locoregional or distant metastasis. While our data do not provide conclusive evidence, these findings endorse the application of minimally invasive approaches and parenchyma-sparing resection or pancreatic resection without lymphadenectomy for small pancreatic neuroendocrine tumors (pNETs) measuring less than 2 cm, when feasible. This recommendation is based on their benign nature and minimal risk of local recurrence. However, in tumors larger than 2 cm, regional lymphadenectomy along the hepatic, splenic, and gastroduodenal ligament should be carried out. Whether there is a difference between an oncological pancreatic resection or a parenchyma-sparing pancreatic resection with lymphadenectomy remains to be investigated.

Several limitations warrant careful consideration when interpreting our results. The current data were collected from 20 clinical cancer registries in Germany; therefore, inconsistency in patient selection, surgical expertise, pathological evaluation, and reporting practices cannot be excluded. Moreover, the registry data did not include detailed information regarding scheme specifics, timing, and protocol completion specific to pNETs. Additionally, certain variables were not part of the available dataset, such as the Ki67% and the number of lymph nodes harvested during surgery; therefore, our analysis could not account for potential confounding factors. Notably, the dataset lacked data pertaining to whether a parenchyma-sparing procedure was performed. Despite these limitations, the large sample size provides analytic power, and the patient cohort is a representation of the general surgical practice in Germany and not only highly specialized centers. To our knowledge, this is the first study to integrate and analyze the effect of LNM on survival outcomes in a simple combined model by integrating tumor size and grade as well as tumor location with LNM. Our results provide data to support clinical decision-making regarding lymphadenectomy in non-metastatic non-functional pancreatic neuroendocrine tumors.

## 5. Conclusions

LNM is a negative independent prognostic factor for DFS in non-metastatic pNETs. Due to the low risk of LNM and recurrence in T1 G1 pNETs, a parenchyma-sparing pancreatic resection without regional lymphadenectomy seems oncologically adequate. In T2 or G2/G3 tumors, especially those located in the pancreatic head, a high prevalence of LNM as well as increased local and distant recurrence in lymph node-positive disease advocate regional lymphadenectomy in addition to pancreatic resection.

## Figures and Tables

**Figure 1 cancers-16-00440-f001:**
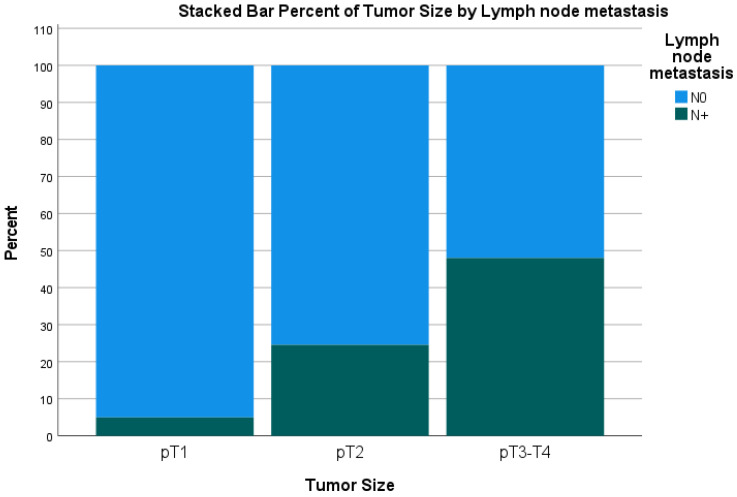
Incidence of lymph node metastasis in pNETs according to tumor size. Legend: N0, no lymph node metastasis; LNM, lymph node metastasis. LNM is 5% in T1, 25% in T2, and 49% in T3–T4.

**Figure 2 cancers-16-00440-f002:**
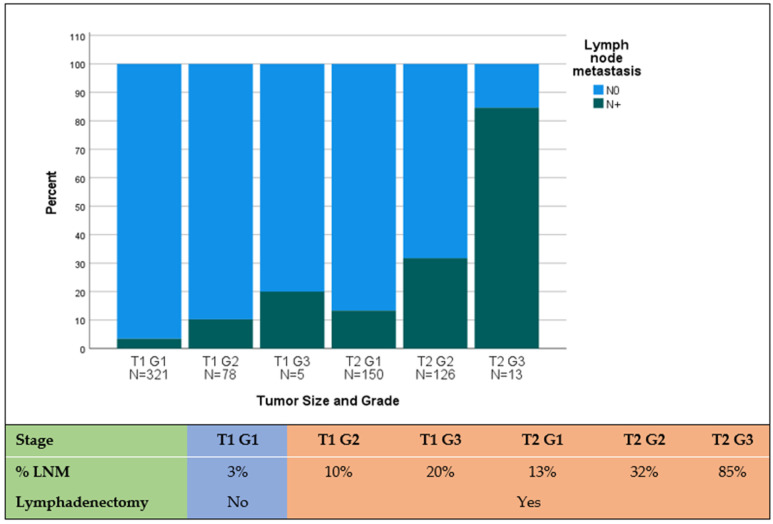
Association of lymph node metastasis with tumor size in combination with tumor grade in pNETs. Legend: N0, no lymph node metastasis; LNM, lymph node metastasis; Orange, indication for lymphadenectomy; Blue, no lymphadenectomy needed.

**Figure 3 cancers-16-00440-f003:**
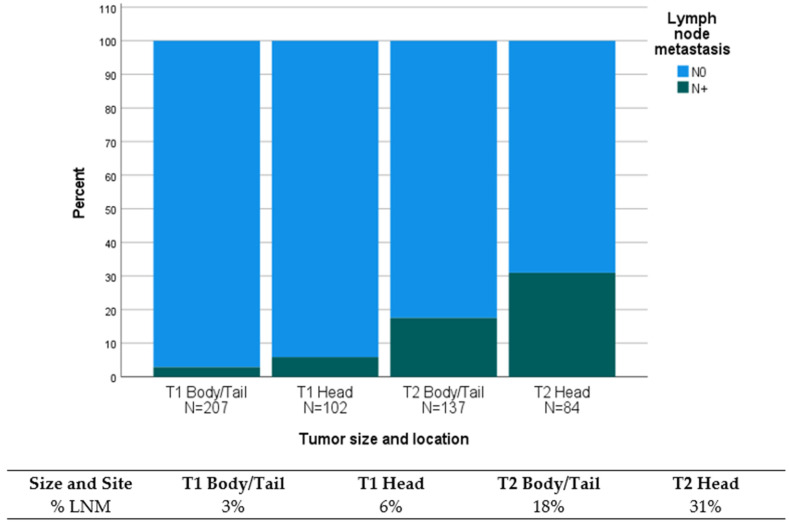
Association of lymph node metastasis with tumor location in T1 and T2 pNETs.

**Figure 4 cancers-16-00440-f004:**
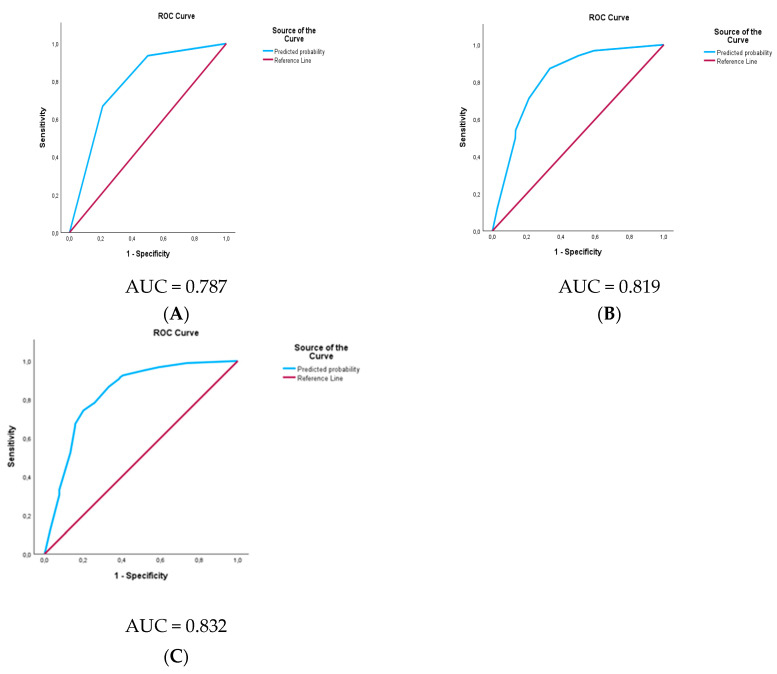
ROC analysis for the combination of tumor size and grade for the prediction of lymph node metastasis. Legend: Area under the curve for lymph node metastasis. (**A**) According to the tumor size only, (**B**) according to the tumor size and grading, and (**C**) according to the tumor size, grading, and location.

**Table 1 cancers-16-00440-t001:** Patient characteristics and association of different variables with the presence of lymph node metastases (LNM) in pNETs.

Variable	N0	LNM	*p*
Age			0.067
<65	366 (73%)	135 (27%)	
≥65	394 (82%)	111 (18%)	
Sex			0.676
Female	356 (75%)	119 (25%)	
Male	404 (76%)	127 (24%)	
Tumor Size			<0.001
T1	384 (95%)	20 (5%)	
T2	218 (75%)	71 (25%)	
T3–T4	158 (51%)	155(49%)	
Tumor Grade			<0.001
G1	497 (87%)	75 (13%)	
G2	237 (65%)	87 (35%)	
G3	26 (39%)	27 (61%)	
Resection margin			0.005
R negative	724 (77%)	208 (22%)	
R positive	36 (49%)	38 (51%)	
Location			<0.001
Head	207 (34.5%)	106 (32.4%)	
Body/Tail	386 (84.7%)	81 (15.3%)	
Local recurrence	23 (3%)	18 (7%)	0.003
Distant metastasis *	67 (11%)	69 (35%)	<0.001

Legend: *p* indicates significance according to the χ^2^ test when comparing patients with and without lymph node metastasis (LNM). * Metastasis in the course of disease.

**Table 2 cancers-16-00440-t002:** Multivariable analysis for disease-free survival.

Variable	Multivariable Analysis		
	HR	95% Cl	*p*
Age, <65 vs. ≥65	1.07	0.74–1.55	0.608
Sex, male vs. female	1.05	0.72–1.54	0.789
Grading			
G2 vs. G1	2.15	1.40–3.30	<0.001
G3 vs. G1	3.45	1.95–6.66	<0.001
Tumor location, head vs. body/tail	1.29	0.89–1.88	0–176
LNM vs. N0	1.79	1.21–2.64	0.003
Tumor size			
T2 vs. T1	2.09	1.10–3.98	0.024
T3–T4 vs. T1	3.45	1.95–6.09	<0.001

Legend: *p* according to Cox regression analysis comparing the specified variables. HR indicates hazard ratio.

**Table 3 cancers-16-00440-t003:** Association of clinicopathologic parameters to lymph node metastasis in pNETs.

Variable	N0	LNM	*p*
Tumor Grade			<0.001
G1	497 (87%)	75 (13%)	
G2	237 (65%)	130 (35%)	
G3	26 (39%)	41 (61%)	
Location			<0.001
Head	207 (66%)	106 (34%)	
Body/Tail	386 (83%)	81 (17%)	
Tumor Size			<0.001
T1	384 (95%)	20(5%)	
T2	218 (66%)	71 (35%)	
T3–T4	158 (51%)	155 (49%)	
Local recurrence			0.022
No recurrence	625 (98%)	186 (94%)	
Recurrence	15 (2%)	11 (6%)	
Distant metastasis (Progression)			<0.001
No distant metastasis	426 (88%)	86 (55%)	
Distant metastasis	60 (12%)	69 (45%)	

Legend: *p* according to the χ^2^ test when comparing patients with and without (N0) lymph node metastasis (LNM).

**Table 4 cancers-16-00440-t004:** Multivariable analysis for prediction of lymph node metastasis in pNETs.

Variable	Multivariable Analysis		
	OR	95% Cl	*p*
Age, <65 vs. ≥65	0.79	0.53–1.16	0.232
Sex, male vs. female	1.14	0.78–1.70	0.498
Resection margin, positive vs. negative	3.07	1.56–6.05	<0.001
Grading,			
G2 vs. G1	2.23	1.45–3.43	<0.001
G3 vs. G1	3.65	1.79–7.32	<0.001
Tumor location, Head vs. Body/Tail	1.87	1.26–2.77	0.002
Tumor size,			
T2 vs. T1	5.31	2.69–10.59	<0.001
T3–T4 vs. T1	14.84	7.67–28.72	<0.001

Legend: *p* according to binary logistic regression analysis comparing the specified variables. HR indicates hazard ratio.

## Data Availability

Data were obtained from the German Cancer Registry Group of the Society of German Tumor Centers and are available upon request from and under the regulations of the Society of German Tumor Centers.

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
