# Peer review of "When Should Lymphadenectomy Be Performed in Non-Metastatic Pancreatic Neuroendocrine Tumors? A Population-Based Analysis of the German Clinical Cancer Registry Group"

_cancers, 2024, doi:10.3390/cancers16020440_

Round 1
Reviewer 1 Report
Comments and Suggestions for Authors
The authors have performed an analysis of data retrieved from the german registry aiming to identify predictive factors of lymph node metastases in patients with non-functioning pNETs. Their findings are very interesting with potential clinical application. However, minor changes are required to render the results clear to the readers.
- In the methods section, please define the classification system for tumor grade.
- In the results section, page 5: 'The 3-,5- and 10- overall survival in nodal negative tumors compared to nodal positive was 84% vs. 79%, 80% vs. 68%, and 56% vs. 52%.' Please provide the statistical significance of the difference. The same applies for the DFS.
- In the section 'Factors associated with lymph node metastasis': As this section is the main topic of the study, I think it should be expanded. It would be better to provide as a table in the main manuscript the Table S1.
Page 6: T2 tumors were mostly G2 44%. Please add brackets.
The statement 'Statistically, LNM was associated with tumors located in the pancreatic head (p < 0.001), positive resection margin (p < 0.001), tumors larger than 2 cm (p < 0.001) and higher tumor grade (G2 vs. G1 and G3 vs G1, p < 0.001).' is different from the multivariable analysis presented below?
Comments on the Quality of English LanguageMinor editing of english language is required.
Author Response
- In the methods section, please define the classification system for tumor grade.
We appreciate your comment. The grading was conducted by pathologists from various institutions and coded based on the ICDO3 system. However, the specific method used, such as Ki67%, Mitotic Index, or morphologic features, was not included in our database. We have acknowledged this as a limitation and have since updated our records to include this information.
- In the results section, page 5: 'The 3-,5- and 10- overall survival in nodal negative tumors compared to nodal positive was 84% vs. 79%, 80% vs. 68%, and 56% vs. 52%.' Please provide the statistical significance of the difference. The same applies for the DFS.
As requested, we added the p-values according to Logrank test respectively.
-In the section 'Factors associated with lymph node metastasis': As this section is the main topic of the study, I think it should be expanded. It would be better to provide as a table in the main manuscript the Table S1.
Thank you for your remark. This is now highlighted as Table 1. Furthermore we expanded the results and included a lymphnode metastasis model (Table S1) and added this to our discussion.
-Page 6: T2 tumors were mostly G2 44%. Please add brackets.
We corrected the text as requested.
-The statement 'Statistically, LNM was associated with tumors located in the pancreatic head (p < 0.001), positive resection margin (p < 0.001), tumors larger than 2 cm (p < 0.001) and higher tumor grade (G2 vs. G1 and G3 vs G1, p < 0.001).' is different from the multivariable analysis presented below?
Indeed, we corrected the discrepancy between the main text and the Table 4. Therefore, we changed the p-value of tumor location accordingly. (p-value<0.001 into p-value 0.002).
Reviewer 2 Report
Comments and Suggestions for Authors
Author Response
Comments to Reviewer 2
-This study showed that lymph node metastasis in upfront resected non-functional non-metastatic pNET was less common in three factors: T1 and G1 classifications, as well as pancreatic body and tail lesions. However, the conclusion does not include one of the factors, tumor localization. It should be added to the conclusion.
Thank you for your remark, the text was changed to:
(…LNM is a negative independent prognostic factor for DFS in non-metastatic pNET. Due to the low risk of LNM and recurrence in T1 G1 pNET, pancreatic resection without regional lymphadenectomy seems oncologically adequate. In T2 or G2/G3 tumors, especially those located in the pancreatic head, a high prevalence of LNM as well as increased local and distant recurrence in lymph node-positive disease advocate regional lymphadenectomy in addition to pancreatic resection…)
-There is no clear evidence whether parenchymal-sparing surgery is superior to conventional resection in terms of pancreatic function, and the grade of the tumor is unknown based on size alone, and even in the same tumor, the grading of the tumor is not clear. In addition, the grading of the tumor is reported to be heterogenous based on size alone, which complicates definitive preoperative grading. Thus the results of this study may provide useful information for predicting lymph node metastasis, but they do not provide evidence for recommending parenchymal-sparing surgery. This limitation should be incorporated into the discussion, and the conclusion should be changed.
We agree that parenchyma-sparing procedures are associated with different perioperative comorbidity and with different resection margin status, since our data does not have information regarding parenchyma-sparing procedures, a recommendation to perform them in T1 G1 tumors cannot be provided, therefore we removed this aspect from our conclusion as stated above. Nevertheless, we can assume that due to the very low incidence of LNM in T1G1 of <5%, the performance of lymphadenectomy in those tumors can be abandoned. Furthermore we added this point into the limitations part.
-The discussion states, "In small pNETs (<2 cm), the benign biology of these tumors and low incidence of local recurrence support the use of minimally invasive procedures and parenchymal-sparing resection when possible.” However, there is a lack of association between minimally invasive surgery and lymph node dissection, which is inappropriate.
Thank you for your response. we corrected the text according to above discussion.
(….While our data does not provide conclusive evidence, these findings endorse the application of minimally invasive approaches and parenchyma-sparing resection or pancreatic resection without lymphadenectomy for small pancreatic neuroendocrine tumors (pNETs) measuring less than 2cm, when feasible. This recommendation is based on their benign nature and minimal risk of local recurrence…).

Reviewer 3 Report
Comments and Suggestions for Authors
General remarks
Large patient group with pNET from a national data set.
The study aim was to identify pre-operative factors associated with lymphnode metastases in pNET. However, no pre-operative factors were assessed, since no information on imaging was included in the study. Only post-operative data were used, which is biased and seems to be incomplete in many items.
The clinical benefit of predicting lymphnode metastases is also unclear, since the survival of patients Is possibly more influenced by for example tumor grade then N-status.; as is shown in the overall survival not being influenced by lln metastases. The disease free survival used seems not a very well validated outcome measure, since no data on follow-up, lenght of follow-up and missing data is included.
Specific remarks;
Methods;
- No information on reliability of the pathological examination of for example tumor grade, which WHO staging system was used in 2010-2021 and how was the data validated.
Results
- 1006 patients included and only 780 patients had a known tumor location in the pancreas and 184 patients had an unknown type of resection. The reliability of the data base is hereby not very strong. Especially information concerning for example resection margin or recurrence could therefore be under estimated.
- Since pre-operative nodal status is unknown and indication for surgery is unclear, the post-operative comparison between patients with N1 and N0 disease is not easy to assess; The fact that survival of patients with N1 disease is worse is not new. The aim of the study was to assess factors associated with N1 disease, why is this information lacking?
- It Is unclear what the multivariable analysis for survival or for risk of lln metastases brings as new information. No new information is revealed, no predictor is useful because operation strategy will not be changed and information Is lacking on clinical effect.
- The AUC curve on location versus risk of N1 disease should not be included sins only 75% of patients had a known tumor location
Overall a study with a lot of data, but with a high bias risk because of the missing data and unknown information on for example follow-up, pathology reporting and missing validation of th edata.
BThis large dataset can only be used to present epidemiological data, the value of the statistical analysis is not high and therefore the clinical proposal for operative strategies in the discussion should not be made.,
-
Comments on the Quality of English Language
The English language used is of good quality
Author Response
Comments to Reviewer 3
-The study aim was to identify pre-operative factors associated with lymphnode metastases in pNET. However, no pre-operative factors were assessed, since no information on imaging was included in the study. Only post-operative data were used, which is biased and seems to be incomplete in many items.
Thank you for this remark. Nowadays, imaging studies represent the most important tool to assess patients with pNET. Tumor size can be very accurately preoperatively defined and correlates with histology. That is why watch and wait strategies which has been advocated by ENETS and these strategies depend solely on imaging. Also tumor location can be accurately preoperatively defined. Therefore, although we cannot prove it, but it is very unlikely that these parameters where not similar in the postoperative histology.
With regards to tumor grading, here, experienced endoscopists and pathologists are required to assess the tumors. This can be challenging but its is possible as provided by Javed et. al. (Annals of Surgery 2022, PMID: 35081574) with increasing success later on the learning curve.
According to this and on the fact that our multivariable analysis showed that these three parameters were independently associated with LNM we constructed our prediction model. This model has been now expanded with a risk-model for LNM added to the supplementary materials (Table S1). Furthermore, we added this information (using of postoperative histology parameters) in the results part (Preoperative prediction of lymph node metastasis).
(… These three parameters can potentially be identified prior to the surgical procedure by cross-sectional imaging and biopsy, and can offer valuable information regarding N+. We created a prediction model with the aim to use these preoperative identifiable parameters, which however, originated from the postoperative histology in our cohort…)
-No information on reliability of the pathological examination of for example tumor grade, which WHO staging system was used in 2010-2021 and how was the data validated.
I appreciate your comment. The grading information was provided from pathologists at various institutions across Germany and was encoded according to the ICDO3 system. However, the specific method employed, whether it be Ki67%, Mitotic Index, or morphologic features, was not documented in our database. We have acknowledged and included this limitation in our study.
- 1006 patients included and only 780 patients had a known tumor location in the pancreas and 184 patients had an unknown type of resection. The reliability of the data base is hereby not very strong. Especially information concerning for example resection margin or recurrence could therefore be under estimated.
Thank you for your comment. Although, specific details about the type of pancreatic resection were not specified for 184 patients. Our inclusion criteria focused on patients with accessible information on pT, pN, pM, Tumor Grade, and R status. Consequently, the data offers dependable insights into disease relapse or progression and resection margin.
- Since pre-operative nodal status is unknown and indication for surgery is unclear, the post-operative comparison between patients with N1 and N0 disease is not easy to assess; The fact that survival of patients with N1 disease is worse is not new. The aim of the study was to assess factors associated with N1 disease, why is this information lacking?
Thank you for this remark. As we explained in our discussion, the importance of LNM is still controversial in the available studies due to different cohort sizes and different follow-up times. Here we provide a large dataset for pNET patients without distant metastasis (n=1006) and long follow-up times (OS median= 30 months and DFS median= 7 months). We also assessed the factors associated with LNM, these are provided in (Table 1).
- It Is unclear what the multivariable analysis for survival or for risk of lln metastases brings as new information. No new information is revealed, no predictor is useful because operation strategy will not be changed and information Is lacking on clinical effect.
Thank you for sharing your thoughts. As previously discussed in point 1, the objective of our multivariable analysis was to determine whether LNM serves as a prognostic factor for DFS. Given that LNM indeed emerged as prognostic for DFS, it underscores the necessity of lymphadenectomy to eliminate positive nodes. Identifying the factors associated with LNM becomes imperative, and the multivariable analysis served as a crucial foundation for constructing our prediction model.
Moreover, in the current era of robotic and minimally invasive surgery, the landscape of pNET treatment has evolved. Surgeons are increasingly opting for parenchyma-sparing procedures, such as enucleation without lymphadenectomy, and shifting away from formal resections to preserve both endo- and exocrine functions (Bolm et al. annals of surgery 2022 PMID: 35758433) Theoretically, this shift may be linked to the possibility of leaving metastatic lymph nodes in situ (since preoperative imaging is still not sensitive enough to assess lymphnodes preoperatively) potentially leading to early tumor relapse. To address this concern, we presented real-world data highlighting the risk of LNM in pNET. For instance, T2 tumors exhibit a 25% incidence of LNM, indicating that in this subgroup, formal resection with lymphadenectomy should be considered.
- The AUC curve on location versus risk of N1 disease should not be included sin only 75% of patients had a known tumor location
Thank you for this remark. We performed a new AUC analysis in patients with complete information regarding, tumor size, tumor grade and location (n=780). We corrected the text and figures accordingly.